# RNA Therapeutics: A Healthcare Paradigm Shift

**DOI:** 10.3390/biomedicines11051275

**Published:** 2023-04-25

**Authors:** Sarfaraz K. Niazi

**Affiliations:** College of Pharmacy, University of Illinois, Chicago, IL 60612, USA; niazi@niazi.com; Tel.: +1-312-297-0000

**Keywords:** mRNA, ribonucleic acid (RNA), vaccines, PCR, autoimmune disorders, therapeutic proteins, repurposing drugs, affordable therapies

## Abstract

COVID-19 brought about the mRNA vaccine and a paradigm shift to a new mode of treating and preventing diseases. Synthetic RNA products are a low-cost solution based on a novel method of using nucleosides to act as an innate medicine factory with unlimited therapeutic possibilities. In addition to the common perception of vaccines preventing infections, the newer applications of RNA therapies include preventing autoimmune disorders, such as diabetes, Parkinson’s disease, Alzheimer’s disease, and Down syndrome; now, we can deliver monoclonal antibodies, hormones, cytokines, and other complex proteins, reducing the manufacturing hurdles associated with these products. Newer PCR technology removes the need for the bacterial expression of DNA, making mRNA a truly synthetic product. AI-driven product design expands the applications of mRNA technology to repurpose therapeutic proteins and test their safety and efficacy quickly. As the industry focuses on mRNA, many novel opportunities will arise, as hundreds of products under development will bring new perspectives based on this significant paradigm shift—finding newer solutions to existing challenges in healthcare.

## 1. Introduction

RNA technology has existed for decades [1]. However, its validation came with the fast spread of COVID-19 [2], which usually takes years to achieve [3]. While several RNA products had been approved, a vaccine that delivers an antigen has resulted in a new focus on future therapies, from the creation of vaccines against infections to preventing or treating autoimmune disorders, supplying proteins by creating an innate manufacturing facility, assisting in gene editing, and many more novel applications [4]. Technological and regulatory advancements include more efficient and lower-cost GMP manufacturing, cell-free DNA sourcing, improved purity profiles, and validated analytical methods from the United States Pharmacopoeia. The realization that we can now treat thousands of untreatable diseases represents a paradigm shift that will bring many ways to utilize nucleoside medicines. In this paper, I provide a perspective on RNA technology and its applications, including examples of proposed mRNA product designs and regulatory plans to assist and encourage developers to achieve faster regulatory approval.

## 2. Understanding mRNA

There are several types and functions of RNAs. There are two types of RNA in cells: messenger RNA (mRNA) and non-coding RNAs such as ncRNA, miRNA, and lncRNA (www.noncode.org). Only 2% of the RNA produced by the human genome encodes proteins. The mRNA is simply a complementary copy of DNA that serves as a template to produce proteins, since it carries information about the amino acid sequence that will make up the target protein. In addition, each mRNA molecule contains non-coding or untranslated regions that regulate the mRNA processing and reading processes (Figure 1).

RNA therapeutics are a class of drug based on ribonucleic acid (RNA) that includes RNA aptamers, messenger RNA (mRNA), RNA interference (RNAi), and antisense RNA (asRNA). The synthesis of proteins is triggered within cells, making it particularly useful in vaccine development, as demonstrated by the recent success of vaccines against the SARS-CoV-2 viral pandemic. Antisense RNA is complementary to coding mRNA. Therefore, it triggers mRNA inactivation to prevent it from being used in protein translation; RNAi-based systems use a similar mechanism involving small interfering RNA (siRNA) and micro-RNA (miRNA). RNA aptamers are short, single-stranded RNA molecules that bind to biomolecular targets and alter in vivo activity.

RNA treats diseases by complementing the miRNA’s active chains, enhancing the function of endogenous miRNA, and reducing intracellular protein expression, blocking miRNA agonists to enhance the expression of a relevant protein. RNA therapy commonly works by utilizing the intracellular enzyme RNase H, or by forming an RNA-induced silencing complex (RISC).

Applications of mRNA include vaccines: (e.g., for SARS-CoV2); individualized medicine (e.g., cancer immunotherapy); cellular treatments; the replacement of proteins; gene editing (e.g., CRISPR CAS9); and treating rare illnesses (e.g., enzyme replacement).

### 2.1. RNA Modifications

Chemical modifications in mRNA are vital for many cell life processes, such as pre-mRNA splicing, nuclear export, transcript stability, and translation initiation (Figure 2). In the 1950s, the first RNA nucleoside modification was identified [6], beginning with the 5′ cap and the poly(A) tail; once technology evolved, many modifications were recognized (Figure 2), revealing that mRNA modifications were not limited to N6-methyladenosine (m^6^A), N1-methyladenosine (m^1^A), 5-methylcytosine (m^5^C), 5-hydroxymethylcytosine (hm^5^C), pseudouridine (Ψ), inosine (I), uridine (U), and ribose-methylation (2′-O-Me). m^6^A is the most abundant modification and has therefore been thoroughly investigated [7,8,9,10].

The value of modifications was demonstrated with the arrival of COVID-19 vaccines, wherein the Ψ modification in the mRNA sequence was essential to ensuring efficacy [11]. One company that did not make this modification ended up with a failed vaccine; however, the cause of this is still debated [12].

Analogous to mRNA modification, many modifications are found in transfer RNAs (tRNAs) and ribosomal RNAs (rRNAs) [13]. Although eukaryotic tRNAs contain, on average, over ten modifications per molecule, from elementary isomerization or methylation to complicated modifications of ring structures, their number of tRNA modifications is the largest and they have the widest chemical variety. Moreover, there are over 200 modifications on human rRNAs.

### 2.2. Self-Amplifying mRNA

RNA developments include self-amplifying RNAs (saRNAs) [14], circular RNAs, and the IRES or RNAs with an internal ribosomal entry site [15].

It is now possible to synthesize saRNAs using in vitro transcription [16] (as conventional mRNA constructs). Still, these are usually larger (9–12 kb) than non-amplifying mRNAs [17]. This has proven to be a promising design capable of delivering larger doses of RNA-translated products that can be more useful for non-vaccine products [18]. Still, it has a disadvantage in that the poly(A) tail cannot be inserted in the expression, and is added in the IVT [19].

Self-amplifying messenger RNAs (saRNAs) utilize the self-replication base of an RNA alphavirus, which may increase RNA transcripts in the cytoplasm, but substitute the viral structural coding genes with genes of interest. Because saRNA transcript replication prolongs expression kinetics, decreasing the frequency of enzyme replacement therapy administration would be advantageous. This method also improves protein expression, requiring 10 times less RNA than linearly modified mRNA to obtain the same protein expression level. Self-amplifying RNA derives its backbone sequence from the alphavirus, a highly replicative positive-sense single-stranded RNA virus. Such mRNA vaccines have an antigen-encoding sequence, a viral RNA polymerase-dependent RNA-encoding sequence, and other components needed for replication. The advantage of the self-replicating approach is that it permits the expression of many more antigens with lower mRNA concentrations. Both RNA vaccines degrade after transient antigen production; however, self-replicating RNA produces prolonged antigen expression [20].

### 2.3. Circular mRNA

The loose ends of RNAs are backfolded during processing to protect circular mRNA (circRNA) against exonuclease activity, improving protein production without increasing the amplitude of protein expression compared to linear modified mRNA. Significantly, by including internal ribosome entry site (IRES) sequences, circRNA eliminates the requirement for expensive 5′ capping and the tedious 3′ poly(A) tail. In addition, circularization lowers the recognition of RIG-1 and Toll-like receptors without chemical substitution. In contrast, the total substitution of uridine with methyl pseudouridine abolishes the translation of circRNA entirely [1].

### 2.4. siRNA

Small interfering RNAs (siRNAs) are non-coding RNA duplexes derived from precursor siRNAs. The latter range in size from 30 to more than 100 bp, and are either transcribed or artificially added. Dicer breaks down the precursor siRNA duplex into 20–30 bp-long siRNA with two base overhangs at the 3′ end, which interacts with RISC to generate RNA interference (RNAi). The RISC’s endonuclease argonaute 2 (AGO2) component cleaves the sense strand while leaving the antisense strand intact, allowing the active RISC to find its target mRNA. The phosphodiester backbone of the target mRNA is then cleaved by AGO2. Because the antisense strand is entirely complementary to the coding area of the target mRNA, siRNA blocks the production of the unc-22, unc-54, fem1, and hlh-1 genes in C. elegans, causing RNAi. In addition, they demonstrated that dsRNA is more successful than ssRNA in inducing RNAi and destroying an mRNA target artificially. Patisiran was the first siRNA-based medication to reach the market. It is used to treat polyneuropathy in adults, which is caused by inherited TTR-mediated amyloidosis.

Patisiran is a dsRNA that promotes the degradation of TTR-encoding mRNA via RNAi [21]. Another siRNA medication, Givosiran, treats acute hepatic porphyria. It decreases the amounts of the disease-causing neurotoxic intermediates aminolevulinic acid and porphobilinogen by targeting aminolevulinate synthase mRNA in the liver [22].

The FDA-approved two siRNA therapeutic products are Givosiran, which targets ALAS1 to treat AHP, and Patisiran, which targets polyneuropathy to treat hereditary transthyretin-mediated amyloidosis.

### 2.5. miRNA

An miRNA (micro-RNA) sequence can be fused along with branches of RNA scaffold, just like in siRNA. This allows for more efficient delivery to affected cells. As they are small, single-stranded, non-coding, and comprise 19–25 nucleotides, miRNAs silence target genes by cleaving mRNA molecules and inhibiting their translation. This family of non-coding RNAs is transcribed from genomic DNA (pri-miRNAs) as primary miRNAs. Before exiting the nucleus, they assume a loop structure with intermittent mismatches and are cleaved by Drosha into 70–100 bp precursor miRNAs (pre-miRNAs).

miRNA-based treatments are divided into two categories: miRNA mimics and miRNA inhibitors. The former are single-stranded RNA oligos that are meant to interfere with miRNAs, whereas the latter are double-stranded RNA molecules that imitate miRNAs. Although no miRNA-based products have yet been approved, there are many potential candidates undergoing clinical testing, such as cobomarsen [14], remlarsen [15], MRG-229 [23], (miR-29 mimic pre-clinical stage), and MRG-110 [24], which are miR-92 inhibitors.

Abnormal miRNA expression is linked to a wide range of illnesses. Targeting dysregulated miRNAs with small-molecule medicines has emerged as a promising new treatment option for various human disorders, particularly cancer. Since gene profiles of the same condition can result in differential transcriptional responses, their functional responses are considered more reproducible, making miRNAs important for gene regulation.

### 2.6. lncRNA

lncRNA therapeutics involve customized siRNAs against specific disorders [25,26]. Both miRNA and lncRNA present many future possibilities for basic research, biomarker discovery, and therapeutic applications [16].

### 2.7. Antisense RNA

Antisense RNA is a non-coding, single-stranded RNA that is complementary to a coding mRNA sequence. It prevents mRNA from translating into proteins. The enzyme “Dicer” cleaves double-stranded RNA precursors into 21–26-nucleotide-long RNA species, producing short antisense RNA transcripts within the nucleus. Antisense medications are based on the idea that antisense RNA hybridizes with mRNA and renders it inactive. These medications are short RNA sequences that bind to mRNA and prevent a gene from making the protein it codes for. Antisense medicines are being developed to treat lung cancer, diabetes, and disorders with a high inflammatory component, such as arthritis and asthma.

Antisense RNA is transferred beyond the cell membrane into the cytoplasm and nucleus, using nonviral vectors and virus vectors such as retrovirus, adenovirus, and liposomes. Because of its high transfection effectiveness, viral vector-based delivery is the most advantageous of the many delivery techniques. However, it is difficult to distribute antisense RNA solely to the targeted regions, and there are numerous restrictions to its application due to antisense RNA’s size and stability difficulties. In addition, improving delivery will require chemical modifications and designing new oligonucleotides.

### 2.8. RNAi

Interfering RNA is a short, non-coding RNA that suppresses gene expression during or after translation. The RNAi system was identified when color genes were introduced into petunias, with the theory that it evolved as a means of innate immunity against double-stranded RNA viruses. However, there are no approved RNAi products on the market.

### 2.9. Aptamers

Aptamers are tiny molecules of single-stranded DNA or RNA that are 20–100 nucleotides long or 3–60 kDa in size. Due to their single-stranded nature, Aptamers can generate various secondary structures, such as pseudoknots, stem-loops, and bulges, via intra-strand base-pairing interactions. The secondary structures included in an aptamer combine to form a unique tertiary structure, which determines which target the aptamer will selectively attach to. As a result, aptamers have a high affinity for their targets, with dissociation constants in the pM to nM range. As a result, they can connect to targets that cannot be helped by tiny peptides made via phage display, or antibodies; they can distinguish between conformational isomers and amino acid changes. Furthermore, because aptamers are based on nucleic acids, they may be directly manufactured, removing the need for the cell-based expression and extraction required in antibody manufacturing. RNA aptamers, for example, may produce a wide range of configurations, leading to the conjecture that they are more discriminating with regard to target affinity than DNA aptamers.

## 3. Approved Therapies

The approved nucleic acid (DNA/RNA) therapeutics (Table 1) treat diseases by targeting their genetic blueprints in vivo, unlike targeting proteins, which is a conventional transient approach. The long-term curative effects of nucleic acid therapies are driven by gene inhibition, addition, replacement, or editing. Unlike other treatment techniques, nucleic acid therapies’ efficacy and applicability in the last few decades have depended on delivery technologies that have enhanced stability, facilitated internalization, and boosted target affinity. Nucleic acid therapeutics include antisense oligonucleotides, ligand-modified small interfering RNA conjugates, lipid nanoparticles, and adeno-associated virus vectors.

Non-coding RNA therapies include short oligonucleotides that bind to complementary sequences in endogenous RNA transcripts and change their processing to replace faulty proteins or existing vaccine antigens. ASO stands for antisense oligonucleotide, a steric block that can prevent polyadenylation, impede or promote translation, or change splicing by physically inhibiting or preventing translation or splicing.

A new type of antisense RNA, splice-switching oligonucleotides (SSOs), can modify gene expression by fixing abnormal splicing, unlike siRNAs, which suppress protein expression. However, it is often difficult to evaluate the effectiveness of SSOs.

The capacity of ASOs to interact with pre-mRNA allows them to target splicing processes. In addition, it dramatically expands the number of RNA sequences that can be selected for ASO binding, reducing off-target effects. For example, only 7% of the 2842 known single-nucleotide polymorphisms in the HTT gene, which codes for the huntingtin protein, can be targeted in mature mRNA (using siRNAs). Still, PCR can target these single-nucleotide polymorphisms 100% of the time.

siRNA stands for small interfering RNA. Small interfering RNAs (siRNAs) are double-stranded RNA molecules that use the RNA-induced silencing complex (RISC) to silence genes. They are 19–23 base pairs long (with a two-nucleotide 3′ overhang). The RISC complex binds siRNA, which is then unraveled via ATP hydrolysis and guided by the enzyme “Slicer” to target mRNA breakdown based on complementary base pairing. As a therapeutic product, siRNA can be delivered through the eye or nose, enhancing bioavailability. However, because intravenous injections require significant amounts, around 20–30% of the total blood volume, targeted distribution to treat malignancies is difficult. In direct tissue/organ electroporation, conjugation to membrane-permeable peptides, and liposome packing in vivo, exogenous siRNAs persist for a few days (at most, a few weeks in non-dividing cells). However, they can use the RISC system to control gene expression by base-pairing to mRNA targets and promoting their destruction when they reach their target.

Table 1 lists the approved oligonucleotides.

### 3.1. “Biosimilar” mRNA Products

mRNA products are classified as biological drugs, despite being synthetic products. However, the FDA has stated that an approved vaccine may not require extensive testing [17]. This observation has led to the concept of modified biological application wherein fewer studies are required if a copy of an mRNA product is presented [18,27].

### 3.2. Synthetic Messenger RNA

Synthetic mRNA has proven its safety and efficacy, and its efficacy offers many opportunities in therapeutics, gene therapy, or vaccine applications [28,29,30]. Using mRNA for personalized and more specific targets introduces new therapeutic modalities; for example, in vitro synthesis based on bacteriophage RNA polymerases, such as T7 or SP6 [31], removes cell culturing or extracting proteins using complex purification methods and offers a scalable manufacturing system [32]. The first commercial-scale cell-free GMP mRNA product was introduced in 2023 [33].

### 3.3. mRNA Construct Design

Typical DNA, pre-mRNA, and mRNA sequences are shown in Figure 3 [34,35].

The transcription of modified nucleic acids is another aspect of mRNA design. When introduced exogenously to a cell, mRNA can induce immunogenicity. However, the innate immune system is not activated when naturally occurring, chemically altered nucleosides, such as pseudouridine and 1-methyl pseudouridine, are present. Additionally, ribosomes translate nucleoside-modified mRNA more quickly than untreated mRNA [41,42]. The sequences of the UTRs and the coding region are optimized to improve translation efficiency and mRNA stability. The UTRs of natural mRNAs, such as those of alpha- or beta-globin, are commonly used as the basis for the UTRs of mRNA vaccines. The 5′ cap of the mRNA strand is introduced either during in vitro transcription (IVT), by including a cap analog in the IVT reaction mixture, or after IVT, by using specialized enzymes (e.g., 2′-O-methyltransferase). Co-transcriptional capping gives lower yields than post-transcriptional capping but may be more cost-effective. Polyadenylation, or the addition of a poly(A) tail at the 3′ end of the mRNA strand, is performed enzymatically, since the poly(A) sequence is long, which will make the plasmid unstable. For example, while most tails are about 70–80 units long, the Pfizer COVID-19 vaccine has 110 tails.

### 3.4. Large-Scale mRNA Production

#### 3.4.1. Template DNA Design and Preparation

An oligonucleotide, a cDNA made from RNA, a plasmid construct, or a PCR output can all be used as the DNA template for mRNA in vitro transcription (IVT). The template has the desired sequence and a double-stranded promoter region where RNA polymerase (such as T7 or SP6) can attach to start the synthesis of RNA. Plasmid constructs are readily obtained from CROs, who can provide these quickly. mRNA production starts with the creation of DNA, which is split into RNA; the DNA comes from plasmid DNA, a circular DNA in E. coli. As shown in Figure 3, the plasmid DNA includes several modifications in the translating region; this generally involves nucleoside modification where uridine is replaced with pseudo-uridine, but this can also be carried out in the IVT process. However, if modifications to translated proteins are required, these modifications are made in the plasmid DNA. For example, the coronavirus’s surface protein sequence is modified to prevent it from collapsing (Figure 4).

The translating region may include nucleoside modification, such as replacing uridine with pseudo-uridine to reduce the immune response to exogenous mRNA and increases its stability to translate the specified sequence. It is also essential to know that the expressed protein need not be the same as the surface protein of a virus, and some modifications are made; for example, K986P and V987P for the COVID-19 vaccine are used to stabilize the translated protein to keep it from collapsing and being pushed out of the cell.

#### 3.4.2. DNA Template Linearization

DNA linearization is a technique used to convert circular DNA molecules, such as plasmids or viral genomes, into linear DNA fragments. This process involves the cleavage of circular DNA at specific sites to generate linear DNA with defined ends. DNA linearization has various applications in molecular biology research, including DNA sequencing, cloning, gene editing, and gene expression studies.

There are several methods that are commonly used for DNA linearization:Restriction enzyme digestion: Restriction enzymes, also known as restriction endonucleases, such as XbaI, cleave DNA at specific recognition sites. Selecting a restriction enzyme that recognizes a site within the circular DNA molecule makes it possible to generate linear DNA fragments with defined ends. The choice of restriction enzyme depends on the recognition site sequence and the desired DNA fragment size.PCR amplification with primers containing restriction sites: PCR (polymerase chain reaction) can amplify a specific region of the circular DNA using primers containing restriction sites. The resulting PCR product can be digested with the corresponding restriction enzyme to linearize the DNA at the desired site.CRISPR-Cas9 cleavage: CRISPR-Cas9 is a powerful gene editing tool that can be used to cleave DNA at specific target sites. By designing guide RNAs (sgRNAs) that target specific sites on the circular DNA, Cas9 nuclease can create double-stranded breaks, resulting in linear DNA fragments when repaired via cellular DNA repair mechanisms.Chemical cleavage: Certain chemicals, such as hydroxylamine or osmium tetroxide, can cleave DNA at specific sites, resulting in linear DNA fragments. Chemical cleavage methods are less commonly used than restriction enzyme digestion or CRISPR-Cas9 cleavage, but can be useful in specific situations.

After linearization, the resulting DNA fragments can be purified and verified using gel electrophoresis or DNA sequencing methods. Linearized DNA can be used in various downstream applications, such as cloning into other vectors, DNA sequencing, gene expression studies, or gene editing using techniques such as CRISPR-Cas9.

It’s important to note that the choice of linearization method depends on the specific experimental requirements, such as the desired DNA fragment size, specific recognition sites, and the availability of appropriate enzymes or reagents. Nevertheless, the proper linearization of circular DNA is critical in many molecular biology experiments, and should be carefully optimized to ensure accurate and reproducible results [11].

The 5′ cap, 3′ cap, and poly(A) tail are added during the IVT process or after transcription using capping enzymes and poly(A) polymerase [43,44,45].

Most aspects of using chemically modified nucleosides or optimizing the nucleoside composition mix have been shown to reduce dsRNA byproducts more than threefold.

mRNA purification:

Unwanted side products from mRNA in vitro synthesis can include double-stranded (dsRNA), uncapped mRNA, and mRNA fragments in varying proportions. These byproducts may overestimate the total amount of functional mRNA, trigger innate immunity, or drastically impede mRNA translation. For size purification, high-performance liquid chromatography (HPLC) is commonly used. Clinical-grade mRNA for therapeutic applications must be free of impurities from upstream processes.

### 3.5. Final Formulation

Formulating lipid nanoparticle (LNP) mRNA products is a critical step in developing mRNA-based therapeutics, including mRNA vaccines and gene therapies. LNPs are lipid-based nanoparticles that encapsulate mRNA molecules, protect them from degradation, facilitate cellular uptake, and enable efficient mRNA delivery into cells. The formulation of LNP mRNA products involves several key components and steps.

Lipid selection: The choice of lipids is a crucial factor in LNP formulation. Different lipids with varying properties, such as cationic, ionizable, and PEGylated lipids, can form LNPs with distinct characteristics. For example, cationic lipids provide a positive charge to LNPs, enhancing cellular uptake, while ionizable lipids allow for the efficient release of mRNA within cells. In addition, PEGylated lipids can improve the stability and pharmacokinetics of LNPs.

mRNA encapsulation: The mRNA molecule carries the genetic information to be delivered into cells and is encapsulated within LNPs. mRNA is complexed with the lipids through electrostatic interactions, forming a lipid–mRNA complex that protects the mRNA from enzymatic degradation in the extracellular environment.

Formulation optimization: The formulation of LNPs is optimized to achieve desired characteristics, such as particle size, charge, and stability. Particle size is an important parameter that affects cellular uptake, biodistribution, and immune response. The surface charge of LNPs can also be controlled by adjusting the composition and ratio of lipids, which can influence their cellular uptake and intracellular trafficking.

Sterilization and quality control: To ensure product safety, LNPs intended for clinical use must be manufactured under sterile conditions. Sterilization methods, such as filtration or gamma irradiation, may be used to eliminate potential contaminants. In addition, quality control tests, including particle size analysis, encapsulation efficiency, and endotoxin testing, are performed to ensure the quality and consistency of LNP mRNA products.

Storage and stability: LNPs are typically stored at low temperatures, such as −20 °C or −80 °C, to maintain stability and prevent degradation. The stability of LNPs can also be influenced by factors such as lipid composition, pH, and storage conditions, and formulation optimization is necessary to ensure the long-term stability and shelf-life of the LNP mRNA products.

Characterization and validation: LNPs are characterized and validated to ensure their quality, safety, and efficacy. Various analytical techniques, such as dynamic light scattering (DLS), transmission electron microscopy (TEM), and reverse-phase high-performance liquid chromatography (RP-HPLC), may be used to characterize the size, morphology, and encapsulation efficiency of LNPs. In vitro and in vivo studies are also conducted to validate the biological activity and therapeutic potential of LNP mRNA products.

The formulation of LNP mRNA products is a complex and crucial process in developing mRNA-based therapeutics. Optimizing lipid composition, encapsulation efficiency, particle size, and stability is essential to ensuring the efficient mRNA delivery, cellular uptake, and therapeutic efficacy of LNP mRNA products. Rigorous quality control and characterization are also critical to ensuring the safety and effectiveness of LNP mRNA formulations for clinical use [46].

### 3.6. Testing

As a new product class, the testing of RNA therapeutic products is still evolving. However, the tests associated with safety and delivery remain the same as those recommended for mRNA vaccines. The products should be tested for their nanoparticle profiles, encapsulation efficiency, in vitro toxicity due to formulation components, stability, storage condition, and shelf-life confirmation. In addition, cell-based assays can be used to test immune responses.

Serological analyses involve protein concentration and activity. Titration assays indirectly measure potency.

A major advance in the validation of RNA-based products has come from the US Pharmacopoeia; though it applies to mRNA vaccines, most of the proposed methods apply to all types of RNA products. In addition, using a USP method reduces the cost of method validation, as verification is only required for the USP tests [47].

## 4. Examples of mRNA Products

The design of mRNA products depends on their use, the most common being the translation of proteins, either as antigens or to replace them. In all such instances, the structure is based on a general plan, as shown in Table 2.

## 5. Perspectives

Many improvements in the RNA therapeutic field are notable and imminent.

### 5.1. Therapeutics vs. Immunization

Immunization requires minimal protein production, as cell-mediated and antibody-mediated immunity can significantly amplify the antigenic signal. To reach a therapeutic threshold, mRNA therapeutics require a 1000-fold higher protein concentration [48]. In many instances, mRNA therapeutics must engage a specific target pathway, cell, tissue, or organ. This criterion emphasizes the efficiency of absorption in the target cell, which impacts the length of expression and the degree of expression. The tissue bioavailability, the circulatory half-life, and the efficiency with which the lipid-based carrier delivers the drug to the tissue might all be rate-limiting factors. With the exception of the liver, which may be easily reached via intravenous medication, effectively administering drugs to solid organs continues to be complicated. Another significant obstacle is repeated dosing, which is frequently necessary for treating chronic disorders. Chronic dosage will eventually activate innate immunity, even with improved chemical changes in mRNA and sophisticated LNPs; however, this will be accompanied by a concurrent decrease in the expression of therapeutic proteins [49,50]. Despite these remaining challenges, a host of emerging technologies are under development to systematically address them [51,52].

### 5.2. Optimization

The caps, the 5′ and 3′ untranslated regions (UTRs), the open reading frame (ORF), and the polyadenylated (poly(A) tail of an individual mRNA can be optimized to increase protein expression. The 5′ cap analogs and 3′ poly(A) are designed to enhance exonuclease protection and complex ribosome catalysis to maximize mRNA stability and translational efficiency [53]. Optimizing the poly(A) tail length (100–300 nucleotides) is essential for balancing the synthetic capability of a given mRNA [54]. Similarly, improved 5′ cap analogs increase translational capacity and enhance capping efficiency [55]. The 3′ and 5′ UTRs can also be tailored to the desired target cell, enhancing translation efficiency and tissue specificity [56].

Currently, most mRNA products contain a synthetic UTR sequence from α-globin or β-globin; UTR optimization can further improve protein expression several-fold [39]. Future UTR sequences may benefit from careful screening and customization to the target of interest. Because of this, it will be possible to tailor each mRNA to the cell that is being targeted, as well as the disease-induced microenvironment, to increase the amount of protein that is produced from each mRNA transcript. This will be carried out to maximize the amount of protein synthesized from each mRNA transcript [57].

Uridine moieties can significantly increase protein expression after transfection in vitro or in vivo. However, the combinations of various types of chemical changes, carriers, methods of in vivo distribution, and levels of mRNA purity reveal a remarkable diversity of effects, which indicates that additional improvement may be achievable. In addition, because naturally produced mammalian mRNAs are often only partially and heterogeneously changed chemically, more improvements can be made by adopting partial nucleoside replacement [58].

## 6. Delivery Systems

Because mRNA is inherently unstable, it is necessary to have a delivery method that can shield it from being degraded by nucleases while still permitting effective cellular absorption, intracellular release, and protein translation. Most developed mRNA therapies are based on LNPs, first identified over sixty years ago [59]. LNPs have undergone numerous modifications and advancements, culminating in their first clinical application in mRNA delivery [60].

### 6.1. Lipid-Based Delivery

Structural lipids, cholesterol, ionizable cationic lipids, and stealth lipids are the four main constituents of LNPs. Most of the structural lipids that make up LNPs are neutrally charged phospholipids. An LNP’s structure is stabilized, and its properties, such as membrane fluidity, elasticity, and permeability, can be modulated by adding cholesterol in varying proportions [61]. Positively charged cationic lipids are needed to load negatively charged nucleic acids into LNPs. However, they have significant disadvantages, as well. Side effects caused by cationic lipids include cytotoxicity, opsonization with plasma proteins, and reduced transfection efficiency. The spleen and the liver work quickly to remove these lipids from the body. The development of several other ionizable lipids, such as DLin-MC3-DMA (MC3), has led to improvements in efficacy, contributing to the clinical delivery of the siRNA Onpattro in 2018 and its clearance by regulatory authorities [62].

The four main components of LNPs are structural lipids, cholesterol, ionizable cationic lipids, and stealth lipids. Neutrally charged phospholipids make up most of the structural lipids that comprise LNPs [21]. Poly sarcosine-conjugated lipids have been developed as an alternative [63].

### 6.2. Nanotechnology

Nanotechnology has emerged as a promising approach to delivering mRNA molecules, enabling the development of mRNA-based therapeutics such as vaccines and gene therapies. Nanotechnology-based delivery systems, including lipid nanoparticles (LNPs), polymeric nanoparticles, and inorganic nanoparticles, offer unique advantages for mRNA delivery, such as protection from enzymatic degradation, improved cellular uptake, enhanced intracellular trafficking, and the controlled release of mRNA. Here, we discuss the role of nanotechnology in mRNA delivery and its potential applications.

Lipid nanoparticles (LNPs): LNPs are one of the most-used nanocarriers for mRNA delivery. They comprise lipids that can self-assemble into nanoparticles, encapsulating mRNA within their lipid bilayers. LNPs can protect mRNA from enzymatic degradation, facilitate cellular uptake through endocytosis, and promote endosomal escape, allowing mRNA to reach the cytoplasm, where it can be translated into protein. LNPs have been successfully used for the delivery of mRNA in mRNA vaccines against infectious diseases such as COVID-19, as well as for mRNA-based gene therapies for genetic disorders.

Polymeric nanoparticles: Polymeric nanoparticles are another type of nanocarrier that can be used for mRNA delivery. These nanoparticles are typically composed of biocompatible and biodegradable polymers, such as poly(lactic-co-glycolic acid) (PLGA), polyethyleneimine (PEI), and polyethylene glycol (PEG). Polymeric nanoparticles can encapsulate mRNA within their matrix or conjugate mRNA onto their surface, providing protection and stability to mRNA during delivery. Polymeric nanoparticles can also be engineered to have specific surface properties, such as charge and size, to optimize the cellular uptake and intracellular trafficking of mRNA.

Inorganic nanoparticles: Inorganic nanoparticles, such as gold and silica nanoparticles, have also been explored for mRNA delivery. These nanoparticles can be engineered to have unique properties, such as surface charge, size, and shape, which can influence their cellular uptake and intracellular trafficking. In addition, inorganic nanoparticles can be functionalized with ligands or peptides to facilitate mRNA binding and uptake by target cells. However, their potential for clinical translation may be limited due to concerns about their safety and biocompatibility.

Surface modification and targeting: Nanoparticles can be surface-modified with various ligands, peptides, or antibodies to target specific cells or tissues, enhancing the selectivity and efficacy of mRNA delivery. For example, targeting ligands can be conjugated onto the surface of nanoparticles to specifically bind to receptors or proteins on the surface of target cells, facilitating the cellular uptake and intracellular trafficking of mRNA. Surface modification can also be used to control the release kinetics of mRNA from nanoparticles, allowing for the sustained or triggered release of mRNA.

Formulation optimization: The formulation of mRNA-loaded nanoparticles can be optimized to achieve desired characteristics, such as particle size, charge, and stability. Particle size is an important parameter that affects cellular uptake, biodistribution, and immune response. The surface charge of nanoparticles can also be controlled by adjusting the composition and ratio of polymers or lipids, which can influence their cellular uptake and intracellular trafficking.

Regulatory considerations: Regulatory considerations are important for developing nanotechnology-based mRNA delivery systems. Nanoparticle-based mRNA products must meet regulatory safety, efficacy, and quality requirements. Regulatory agencies such as the US Food and Drug Administration (FDA) have specific guidelines for developing and approving mRNA-based therapeutics, including requirements for the manufacturing, characterization, stability, and safety assessment of nanotechnology-based mRNA delivery systems.

In conclusion, nanotechnology-based delivery systems hold great promise for the efficient and targeted delivery of mRNA molecules for various therapeutic applications. LNPs and polymeric and inorganic nanoparticles are among the most widely studied nanocarriers for mRNA delivery. They can protect mRNA from degradation, facilitate cellular uptake, promote endosomal escape, and allow for the controlled release of mRNA. In addition, surface modification and targeting strategies can enhance the selectivity and efficacy of mRNA delivery, while formulation optimization can fine-tune the characteristics of nanoparticles for optimal performance. However, regulatory considerations are crucial to ensure nanotechnology-based mRNA products’ safety, efficacy, and quality.

The use of nanotechnology for mRNA delivery has shown promising results in preclinical and clinical studies. For example, lipid nanoparticles have successfully delivered mRNA in mRNA vaccines against infectious diseases such as COVID-19, demonstrating their high efficacy and safety profiles. mRNA-loaded polymeric nanoparticles have also been investigated for gene therapies, showing potential for treating genetic disorders such as cystic fibrosis and rare genetic diseases. Inorganic nanoparticles have shown promise in preclinical studies for mRNA delivery, although further research is needed to address safety concerns.

Despite these promising results, challenges remain in the field of mRNA delivery using nanotechnology. These challenges include the optimization of nanoparticle formulations for different types of mRNA molecules, achieving efficient delivery to specific target cells or tissues, minimizing the potential toxicity and immunogenicity of nanoparticles, and ensuring the scalability and reproducibility of nanoparticle manufacturing processes. Additionally, the long-term safety of nanotechnology-based mRNA delivery systems must be carefully evaluated through extensive preclinical and clinical studies.

In conclusion, nanotechnology-based delivery systems hold great potential for the effective and targeted delivery of mRNA molecules for various therapeutic applications. Lipid nanoparticles, polymeric nanoparticles, and inorganic nanoparticles are among the most promising nanocarriers for mRNA delivery. Further research and development efforts are needed to overcome challenges and fully realize the potential of nanotechnology in advancing mRNA-based therapeutics. With continued advancements in nanotechnology, mRNA delivery using nanocarriers has the potential to revolutionize the field of medicine and pave the way for developing novel mRNA-based therapies for various diseases.

Synthetic compounds that comprise a few essential components to form complex structures are known as polymer nanomaterials. Synthetic polymers such as PLGA [poly(lactic-co-glycolic acid)], PLA (polylactic acid), chitosan, gelatin, polycaprolactone, and poly alkyl-cyanoacrylates are commonly used in these materials.

MSNPs (mesoporous silica nanoparticles) comprise an amorphous silica (silicon dioxide) matrix with mesoporous ordered porosity. This nanoparticle’s unique qualities include a huge surface area with an enormous pore content, simplicity of customization, and established silanol chemistry. To transport negatively charged RNA, the surface of the nanoparticles is changed by positively charged moieties. As a result, the pore diameters in MSNPs are adjusted across a wide range and are quite consistent. As a result, the particles have a high nucleic acid loading capacity and effective delivery.

Gold nanoparticles, quantum dots, nanographene oxide, and carbon nanotubes are synthetic nanostructures that can hold RNA, protect it from degradation, and deliver it to the disease site of interest.

### 6.3. Cell-Based Delivery

The cell-based delivery of mRNA refers to using living cells as carriers or vehicles to deliver mRNA molecules into target cells for therapeutic purposes. This approach leverages the natural ability of cells to take up and process mRNA, allowing for the efficient and targeted delivery of therapeutic mRNA to specific cells or tissues.

There are several different strategies for the cell-based delivery of mRNA. One common approach is to use specialized cells, such as stem or immune cells, which can be home to specific tissues or organs. These cells can be engineered to express or produce mRNA molecules of interest, which are released and taken up by target cells. For example, stem cells can be genetically modified to produce mRNA that encodes a therapeutic protein, and when administered to a patient, they can migrate to the target tissue and deliver the therapeutic mRNA to nearby cells for protein production.

Another approach is to directly engineer target cells to express exogenous mRNA. This can be achieved through electroporation, where an electrical field is applied to cells to create temporary pores in their cell membranes, allowing for mRNA molecules to enter the cells. Alternatively, mRNA can be directly injected into target cells using microinjection techniques, which involve injecting mRNA molecules into individual cells using specialized equipment.

The cell-based delivery of mRNA has shown great promise in various therapeutic applications. One of the most notable examples is mRNA-loaded immune cells for cancer immunotherapy. Immune cells, such as dendritic or T cells, can be engineered to express tumor-specific mRNA, allowing them to target and stimulate the immune response against cancer cells. This approach has shown promising results in preclinical and clinical studies, with some mRNA-based immunotherapies showing significant anti-cancer activity.

The cell-based delivery of mRNA also holds potential for regenerative medicine and tissue engineering. For example, stem cells can be engineered to express tissue-specific mRNA, which can then be used to direct their differentiation into specific cell types for tissue repair or regeneration. This approach has been explored for various tissues, including the heart, liver, and cartilage, and has shown promising results in preclinical studies.

However, challenges remain in the field of cell-based mRNA delivery. The optimization of cell-based delivery methods, including cell type selection, mRNA modification, and delivery techniques, is still an active area of research. Ensuring the safety and efficacy of cell-based mRNA delivery approaches, including potential immunogenicity and tumorigenicity concerns, also requires careful evaluation in preclinical and clinical studies.

In conclusion, the cell-based delivery of mRNA represents a promising approach to the targeted and efficient delivery of mRNA molecules for therapeutic purposes. It has shown great potential in various fields, including cancer immunotherapy and regenerative medicine. Further research and development efforts are needed to optimize cell-based mRNA delivery methods, address safety concerns, and fully realize the potential of this approach in advancing mRNA-based therapies and extracellular vesicle-based delivery.

### 6.4. Extracellular Vesicles

Extracellular vesicles (EVs) are small membranous structures released by various cells into the extracellular space. They play a crucial role in cell-to-cell communication by transporting biomolecules, including proteins, lipids, and nucleic acids, to neighboring or distant cells. In recent years, EVs have emerged as promising vehicles for mRNA delivery, offering a new approach to gene therapy and other biomedical applications.

mRNA, or messenger RNA, is an RNA molecule that carries the genetic information necessary for protein synthesis. By delivering mRNA to target cells, it is possible to manipulate gene expression and potentially treat a wide range of diseases caused by genetic mutations or other molecular deficiencies. However, delivering mRNA into cells is challenging due to the large size and negative charge of mRNA molecules, as well as the presence of cellular barriers that prevent their efficient uptake.

EVs solve these challenges by serving as natural carriers for mRNA. They are released by cells and can be isolated from various body fluids, such as blood, urine, and saliva, making them a non-invasive and scalable source for mRNA delivery. In addition, EVs comprise a lipid bilayer membrane that protects the encapsulated mRNA from degradation by nucleases and provides stability during circulation in the body.

One of the key advantages of using EVs for mRNA delivery is their ability to target specific cells or tissues. EVs can be derived from different cell types, such as immune cells, stem cells, or cancer cells, and can be engineered to express specific surface proteins or ligands that allow them to bind and enter target cells selectively. This targeted delivery can enhance the efficiency and specificity of mRNA delivery, reducing off-target effects and improving therapeutic outcomes.

Moreover, EVs have shown excellent biocompatibility and low immunogenicity, making them suitable for in vivo applications. They can bypass the reticuloendothelial system (RES) and other immune surveillance mechanisms, which can eliminate or neutralize foreign substances, allowing for prolonged circulation and increased accumulation at the target site. This property makes EVs an attractive option for systemic mRNA delivery, where the mRNA can be delivered to cells throughout the body.

In addition, EVs can protect the mRNA cargo from degradation and improve its stability. The lipid membrane of EVs can shield the mRNA from enzymatic degradation and protect it against harsh environmental conditions. This property can increase the half-life of mRNA and ensure its intact delivery to the target cells, which is critical for achieving the desired therapeutic effect.

Furthermore, EVs could potentially overcome the challenges associated with the intracellular delivery of mRNA. Once inside the target cells, EVs can release their mRNA cargo into the cytoplasm, where the cellular machinery can translate the mRNA into functional proteins. This intracellular delivery mechanism can bypass the need for complex delivery methods, such as transfection or viral vectors, which can be associated with safety concerns and technical limitations.

In summary, EVs offer a promising approach to mRNA delivery, potentially revolutionizing gene therapy and other biomedical applications. Their natural origin, targeting capabilities, biocompatibility, and ability to protect mRNA cargo make them an attractive option for mRNA-based therapeutics. Further research and development in this field will likely unveil the full potential of EVs as mRNA delivery vehicles and pave the way for new therapeutic strategies for a wide range of diseases [64,65].

### 6.5. Biomimetic Delivery

The biomimetic delivery of mRNA is a cutting-edge approach that draws inspiration from nature to develop innovative strategies for delivering mRNA molecules to target cells or tissues. By mimicking the natural processes of cellular uptake and intracellular trafficking, biomimetic mRNA delivery systems aim to overcome the challenges associated with traditional delivery methods and enhance the safety and efficacy of mRNA-based therapies.

Nature provides a wealth of examples that inspire the creation of biomimetic mRNA delivery strategies. For instance, viruses are known for their ability to deliver genetic material into host cells efficiently, but they can also pose safety concerns. Biomimetic delivery systems can replicate the viral mechanisms of cellular entry, but without the risk of viral replication or immunogenicity.

One approach to biomimetic mRNA delivery is to use liposomes or lipid nanoparticles that mimic the natural lipid bilayer of cell membranes. These lipid-based delivery systems can encapsulate mRNA molecules within their lipid bilayers, protecting them from degradation and facilitating cellular uptake. In addition, the liposomes or lipid nanoparticles can be engineered to mimic the surface properties of specific cells, allowing them to selectively bind to target cells and enter them via endocytosis, a natural cellular uptake process. Once inside the cells, the liposomes or lipid nanoparticles can fuse with the endosomal membranes and release the mRNA cargo into the cytoplasm, which can be translated into proteins.

Another biomimetic approach uses cell-derived vesicles, such as exosomes or microvesicles, as mRNA delivery vehicles. Cells naturally release these vesicles, and can be isolated and engineered to carry mRNA molecules. Exosomes, in particular, are small vesicles that are secreted by cells into the extracellular space and are known to play a role in intercellular communication. They have a lipid bilayer membrane and can carry a variety of biomolecules, including mRNA, making them attractive for biomimetic mRNA delivery. In addition, exosomes can be engineered to express specific surface proteins or ligands that allow them to target particular cells or tissues, facilitating their uptake and intracellular release of mRNA.

Biomimetic mRNA delivery systems also leverage cellular pathways for intracellular trafficking to enhance delivery efficiency. For example, some delivery systems are designed to take advantage of endosomal escape pathways, which allow endocytosed materials to escape from endosomes into the cytoplasm. By replicating these pathways, biomimetic mRNA delivery systems can ensure that the mRNA cargo is released into the cytoplasm, where it can be translated into proteins.

One of the advantages of biomimetic mRNA delivery is the potential for enhanced safety and biocompatibility. These systems can be designed to minimize immunogenicity and off-target effects, as they often use natural or naturally derived materials that are well-tolerated by the body. Additionally, biomimetic mRNA delivery systems can offer controlled release of mRNA, allowing for the precise modulation of gene expression levels and minimizing the risk of overexpression or unintended effects.

Biomimetic mRNA delivery holds great promise for various applications, including gene therapy, vaccinations, and regenerative medicine. It has the potential to revolutionize the field of mRNA therapeutics by providing safe, effective, and targeted delivery of mRNA molecules to specific cells or tissues. However, challenges still need to be addressed through solutions such as improving delivery efficiency, optimizing biodistribution, and ensuring the long-term stability of mRNA cargo. Further research and development in this field will likely lead to exciting breakthroughs and new possibilities for mRNA-based therapies.

Biomimetic delivery combines biological and synthetic particles, such as a synthetic core with defined binding properties, to encapsulate the cargo (such as gold, silica, LNPs, or polymers), which is then coated with a cellular membrane [66]. The coating reduces the immunogenicity of synthetic materials (such as an anti-PEG antibody), enables tissue targeting based on the cell source, and increases the stability of particles in circulation.

Combining biological (e.g., EVs) and synthetic (e.g., LNPs) components to form hybrid particles that complement the coating is a biomimetic approach [67]. These hybrid particles retain the biocompatibility and targeting specificity of EVs while possessing the controlled manufacturing and stable storage capacities of LNPs [68]. However, this method is still in its infancy.

### 6.6. Tissue Targeting

For mRNA treatments to reach their full potential, more improved in vivo delivery technologies are required. These systems are especially important for solid organs, including the heart, kidneys, brain, and lungs. Regarding the convenience of delivery, the liver is the organ of choice for most molecular medicines. Its fenestrated vasculature makes it easier to administer homogenous substances effectively and for large particles to move through. Therefore, easy intravenous delivery facilitates the effective production of mRNA cargos in the hepatocytes, which ultimately results in therapeutic protein levels. When the target is not the liver, improved delivery systems are needed, such as administering catheters to the target organ [69] or engineering delivery systems, such as using HSV’s tropism for neural cells to target neurons [70].

### 6.7. Inhalation, Intranasal, and Injection Delivery

In contrast to the liver, the kidney acts as a filter that excludes big molecules while permitting only small ones to pass through. Most molecular therapies transported from the circulation to the kidney are impeded because the glomerulus actively removes proteins with a molecular weight of more than 50 kDa, and constitutive podocytes form slit diaphragms with diameters that are only 10 nm [71]. Adjusting the depth of a needle or catheter’s implantation into the kidney makes it possible to administer a direct subcapsular injection into the medulla or cortex of the kidney.

Through inhalation, it is possible for the molecules to reach the lungs instantly, enabling the use of lower drug dosage and, as a result, reducing the severity of unpleasant systemic side effects. Systems that are attractive for pulmonary delivery allow access to the alveoli and lung parenchyma that is direct, quick, and does not require intrusive procedures. Additionally, because the nuclease activity on the airside of the lung is lower than it is on the other side of the lung, this side of the lung is a more conducive environment for the preservation of RNA [70]. Finally, there are potential avenues for the non-invasive delivery of drugs to the brain, provided by the neuronal networks that connect the nasal mucosa and the brain [72].

### 6.8. Chronic Dosing

For mRNA to successfully move from vaccines to treatments, the ability to distribute mRNA in a targeted and efficient manner on several occasions, while preserving high protein yields, is essential. Enzyme replacement therapies, which are dependent on recombinant proteins, provide a striking illustration of this point. For example, systemic injections of factor VIII or factor IX recombinant proteins are routinely used to treat hemophilia A and B, which are blood diseases characterized by a deficiency in clotting proteins. These injections are often given anywhere from three to seven times each week [73].

### 6.9. Intracellular Delivery

Compared to protein medications, the ability of mRNA therapy to produce high levels of intracellular proteins represents the true added value that these treatments possess. Furthermore, the direct targeting of metabolic illnesses, such as Crigler–Najjar syndrome [74], methylmalonic acidemia, [75] propionic acidemia, and cystic fibrosis, which are difficult-to-treat diseases using proteins [76], is made possible using this intracrine technique.

Moreover, it enables the quick creation of therapeutic antibody mixtures. Considerable potential exists among intracellular proteins, such as metabolic and mitochondrial proteins.

Therapeutic doses of recombinant antibodies are often quite high. It is not apparent whether it is possible to obtain these high concentrations by delivering encoding mRNA. Despite this, several factors suggest that mRNA is better than recombinant antibodies. First, antibody expression in situ can lower the protein required for a therapeutic effect due to the high protein concentration in the local environment. Second, there has not been any evidence of saturation or dose-limiting toxicity for mRNA-mediated antibody delivery based on the doses that have been evaluated until this point. Third, target-specific mRNA optimization and further improvements to the formulation can substantially increase its efficacy [77]. To date, the viability of the mRNA platform for antibody therapy has been examined in the fields of oncology, infectious disorders, and antitoxins [78].

### 6.10. Gene Editing

mRNA can be utilized for gene editing by encoding nucleases. The precision of “cutting” and “pasting” genomic DNA in specific locations is a promising therapeutic area for applying mRNA technology. mRNA translates nucleases such as zinc-finger nucleases (ZFNs), transcription activator effector nucleases (TALENs), and CRISPR-Cas9. These genetic engineering technologies allow for the replacement or adjustment of gene expression by introducing or deleting predefined modifications in the genome of target cells. By joining the non-homologous end (NHEJ) or performing a homology-directed repair or insertion, a target gene can be corrected by deleting disease-causing mutations or adding protective mutations (HDR). ZFNs and TALENs make it easier for proteins to recognize a sequence via protein–DNA interactions. Still, the extensive engineering required to generate DNA recognition and binding domains in proteins limits their use.

## 7. Autoimmune Disorders

When the body’s immune system attacks and destroys body tissue inadvertently but is driven by innate evolutionary mutations, many diseases arise, and almost 80 have been identified. In most cases, the cause is unknown, and the disease destroys body tissue and causes abnormal growth of an organ and changes in organ function, and often, multiple organs are affected. Primary autoimmune disorders include:Addison diseaseAlopecia areataAlzheimer’s diseaseAnkylosing spondylitisCeliac diseaseDementiaDermatomyositisGraves’ diseaseHashimoto’s thyroiditisMultiple sclerosisMyasthenia gravisParkinson’s diseasePemphigusPernicious anemiaPolymyalgia rheumaticaPsoriasisReactive arthritisRheumatoid arthritisSclerodermaSjögren’s syndromeSystemic lupus erythematosusTemporal arteritisType I diabetesVasculitis

The financial impact of these diseases is very high because of their lifetime impact on quality of life; type 1 diabetes is the best example of this. When an immunogenic response is caused by an aberrant antibody, such as in the case of type 1 diabetes, or the synthesis of coating proteins, such as in Alzheimer’s disease, these can be prevented by mRNA therapies that deliver an antigen to produce an antibody that views another rogue antibody or a protein as antigen, permanently removing it. For example, in type 1 diabetes, a protein sequence found in the anti-GAD antibody will produce an antibody that will view the anti-GAD antibody as an antigen.

## 8. Regulatory Status

The regulatory status of RNA-based therapeutics remains unclear. While chemically produced oligonucleotides, such as ASOs and siRNAs, are not part of the ICH Q3A and B guidelines that address the issue of “impurities in new drug substances and in drug products produced by chemical synthesis”, they also do not comply with the ICH Q6A guideline about “specifications: test procedures and acceptance criteria for new drug substances and new drug products: chemical substance.” Meanwhile, mRNA vaccines are not classified as “gene therapy” since they are defined as “a medical intervention based on the modification of the genetic material of living cells”, [79] which applies to these molecules. It is anticipated that regulatory guidelines specific to RNA therapeutic products will be developed as more products are filed for approval [80].

The testing of RNA therapeutic products can better understood by reviewing the tests conducted to assess the quality of ASOs, siRNAs, and mRNAs (see Table 2). This information was developed based on the information from the Committee for Medicinal Products for Human Use (CHMP) public assessment reports of authorized ASOs, siRNAs, and mRNAs written by the EMA [81]. (Table 3)

## 9. Conclusions

RNA therapeutics is a wide field that includes therapies and the prevention of infection and autoimmune disorders. The roles of RNA and its dozens of types are now well recognized, but only a few applications have been introduced. The consistency with which RNA operates at all levels, from translation to its limited lifecycle, makes it an ideal modality for treating and preventing diseases, mostly untreatable ones. Unlike chemical drugs, its toxicity is limited to a localized distribution site. Unlike DNA therapies, it does not enter the nucleus, reducing the risk of any damage to the genes in the nucleus. It is anticipated that RNA therapeutics will be the most forthcoming branch of new drug discovery, and for this reason, a contemporary understanding of RNA developments is essential for developers.

mRNA (messenger RNA) technology has emerged as a promising approach to producing therapeutic proteins, offering several advantages over traditional recombinant technology or in vitro translation methods.

The following are some of the advantages of mRNA technology for producing therapeutic proteins:Rapid development and manufacturing: the mRNA-based production of therapeutic proteins offers a faster and more efficient approach compared to recombinant technology or in vitro translation. Traditional recombinant technology requires cloning and expression in host cells, which can be time-consuming and complex. In contrast, mRNA technology involves synthesizing mRNA molecules in vitro, and then, introducing them into cells for translation. The procedure of mRNA transfection is quicker and more effective. mRNA is directly transferred to and expressed in the cytoplasm and has a smaller build than plasmid DNA, never crossing the nuclear membrane. This allows for the rapid and scalable production of mRNA, making it an attractive option for producing therapeutic proteins with shorter development timelines. Producing therapeutic proteins necessitates extensive cell culture and time-consuming, protein-specific purification procedures.Flexibility and adaptability: mRNA technology offers greater flexibility and adaptability compared to recombinant technology or in vitro translation. mRNA molecules can be easily modified by adding or removing specific sequences, allowing to produce a wide range of therapeutic proteins with different properties. This flexibility makes mRNA technology well-suited to producing complex proteins, including those that are difficult to express using recombinant methods. Additionally, mRNA technology can be easily adapted to produce new proteins in response to changing medical needs, making it a versatile platform for therapeutic protein production.Reduced risk of contamination: Unlike recombinant technology, which involves the use of genetically modified organisms (GMOs) for protein production, mRNA technology does not require the use of living cells. This reduces the risk of contamination with unwanted or harmful substances, such as endotoxins or adventitious agents that may be present in cell-based production systems. This makes mRNA technology a safer option for producing therapeutic proteins, with fewer concerns related to safety and regulatory compliance.Lower manufacturing costs: mRNA technology has the potential to lower the manufacturing costs associated with producing therapeutic proteins compared to recombinant technology or in vitro translation. Traditional recombinant technology often requires extensive downstream processing steps, such as protein purification and refolding, which can be costly and time-consuming. In contrast, mRNA technology eliminates the need for these labor-intensive steps, as the proteins are produced directly from the mRNA molecules inside the cells. No matter the coding sequence, mRNA is generated in a typical one-vessel reaction utilizing the same procedure [82]. Synthetic mRNA can be created, using mRNA technology, to look like molecules that naturally exist in the cytoplasm of cells and transiently deliver the desired proteins into cells [81,83]. This can result in a more cost-effective production process with reduced manufacturing expenses.Enhanced safety profile: mRNA technology offers an improved safety profile compared to traditional recombinant technology or in vitro translation methods. mRNA molecules are non-infectious and do not integrate into the host genome, reducing the risk of unintended genetic modifications or insertional mutagenesis. Additionally, mRNA technology allows for precise control over the expression of therapeutic proteins, reducing the risk of overexpression or off-target effects. Misfolded or poorly changed proteins can have negative effects and can be immunogenic. Plasmid DNA transfection is less effective in dormant cells, and the necessity for a particular promoter and crossing of the nuclear membrane complicates the procedure [84]. This makes the mRNA-based production of therapeutic proteins a safer option with fewer safety concerns.Scalability: mRNA technology offers scalability advantages compared to in vitro translation methods. In vitro translation methods can be limited by the availability of appropriate cell-free systems and may have lower yields. In contrast, mRNA technology can be scaled up to meet larger production demands by simply increasing the amount of mRNA synthesized and introduced into cells. The complexity of the protein synthesis process frequently necessitates lengthy development cycles and makes GMP compliance difficult. This scalability makes mRNA technology suitable for the commercial production of therapeutic proteins on a large scale.

In conclusion, mRNA technology offers several advantages in the production of therapeutic proteins compared to recombinant technology or in vitro translation methods. It offers rapid development and manufacturing, flexibility and adaptability, a reduced risk of contamination, lower manufacturing costs, an enhanced safety profile, and scalability. These advantages make mRNA technology a promising and innovative approach to producing therapeutic proteins, with potential for significant advancements in the field of biopharmaceuticals. Over therapeutic proteins and DNA transfections, mRNA technology has the following advantages:

## Figures and Tables

**Figure 1 biomedicines-11-01275-f001:**
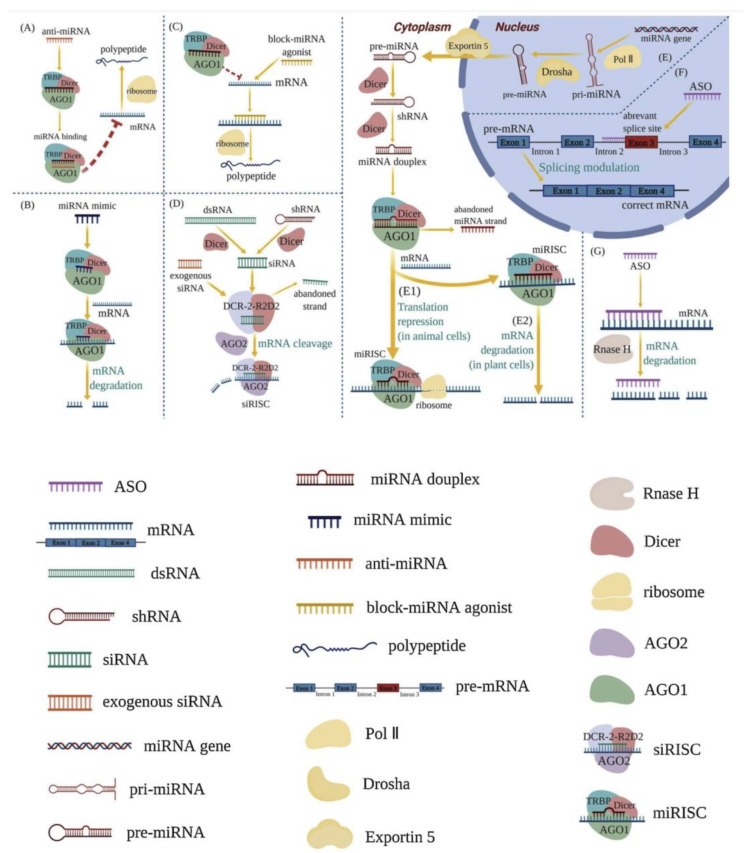
Mechanisms of RNA therapeutics [5] (Via license: CC BY-NC-ND 4.0). anti-miRNA (**A**); miRNA mimics (**B**); block-miRNA agonists (**C**); RNA-induced silencing complex (RISC) (**D**,**E1**,**E2**); splicing process of mRNA (**F**); intracellular enzyme RNase H (**G**). Key: shRNA—short hairpin RNA; pri-miRNA—primary miRNA; pre-miRNA—precursor miRNA; TRBP—Tar RNA binding protein; AGOs—argonautes; Pol II—RNA polymerase II; siRISC—siRNA-induced silencing complex; miRISC—miRNA-induced silencing complex.

**Figure 2 biomedicines-11-01275-f002:**
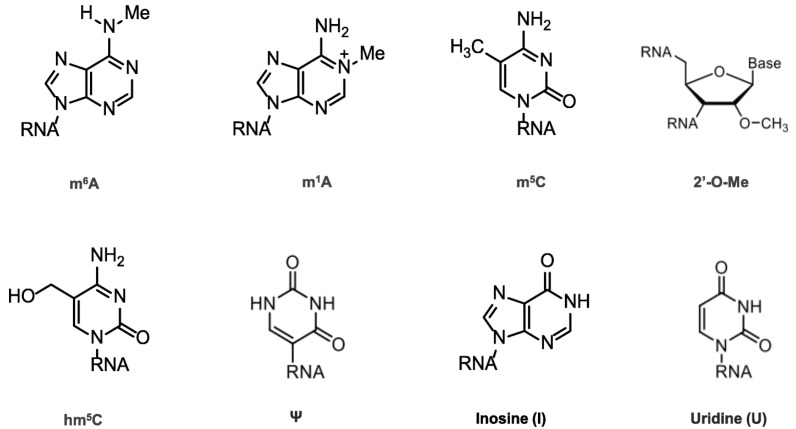
Chemical structures of mRNA modifications. Chemical structures in eukaryotic mRNA, including m^6^A, m^1^A, m^5^C, hm^5^C, Ψ, I, U, and 2′-O-Me. m^6^A is mostly found in the 3′UTRs as well as the 5′UTRs; m^1^A-containing mRNA is 10 times less common than m^6^A-containing mRNA, but it is found in all segments of mRNA; m^5^C is found in both coding and non-coding regions, especially in GC-rich regions, where it regulates transcription differently; Ψ appears in many locations; 2′-O-Me is concentrated in the mRNA region that encodes amino acids.

**Figure 3 biomedicines-11-01275-f003:**
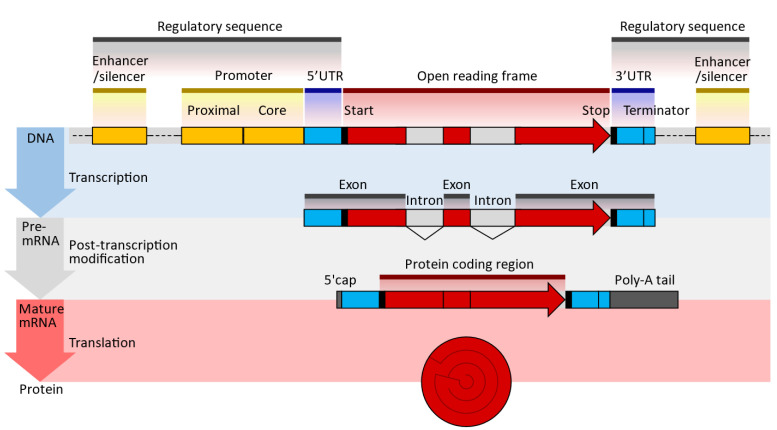
Typical DNA, pre-MRNA, and mRNA design sequences; UTR—untranslated region; poly(A)—polyadenylate signal tail (enumver as beeded [36]). The 5′ cap at the end allows for sequence recognition, protecting the translated molecules from digestion by nucleases [37]. The 5′UTR/3′UTR determines the translation efficiency, stability, and location; it is pivotal to optimizing expression [36,38]. The open reading frame or coding sequence (CDS) lists the genes expressed. These genes are optimized and modified to improve translational efficiencies, such as the modification of guanine and cytosine content [39]. The Poly(A) tail is essential for optimal translation 11 and improves stability by blocking digestion by 3′ exonuclease, increasing translation efficiency and adding to the molecule’s stability [40].

**Figure 4 biomedicines-11-01275-f004:**
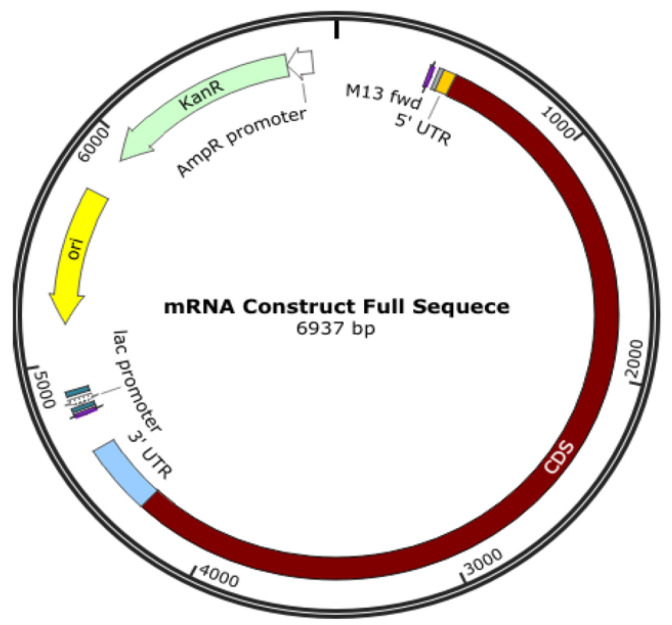
Plasmid DNA is designed to produce mRNA for a COVID-19 vaccine.

**Table 1 biomedicines-11-01275-t001:** Approved oligonucleotides.

Products	Gene Target	Indication	Administration	Approval Year	Cost (USD/Treatment)
ASOs					
Vitravene, fomivirsen (Ionis Pharmaceuticals)	Cytomegalovirus gene (UL123)	Cytomegalovirus infection	Intravitreal	1998 (withdrawn in 2002/2006)	10.4 k/year
Exondys 51, eteplirsen (Sarepta Therapeutics)	Dystrophin (exon 51)	Duchenne muscular dystrophy	Intrathecal	2016	300 k/year
Tegsedi, inotersen (Ionis Pharmaceuticals)	Transthyretin (TTR)	TTR-mediated amyloidosis	Subcutaneous	2018	450 k/year
Spinraza, nusinersen (Ionis Pharmaceuticals)	Survival of motor neuron 2 (SMN2)	Spinal muscular atrophy	Intrathecal	2016	750 k/year, 375 k/year
Kynamro, mipomersen (Ionis Pharmaceuticals)	Apolipoprotein B-100	Hypercholesterolemia	Subcutaneous	2013	176 k/year
Waylivra, Volanesoren (Ionis Pharmaceuticals/Akcea)	Apolipoprotein CIII	Familial chylomicronemia syndrome	Subcutaneous	2019	395 k/year
Vyondys 53, golodirsen (Sarepta Therapeutics)	Dystrophin (exon 53)	Duchenne muscular dystrophy	Subcutaneous	2019 (confirmatory trial required)	300 k/year
Amondys 45, casimersen (Sarepta Therapeutics)	Dystrophin (exon 45)	Duchenne muscular dystrophy	Subcutaneous	2021	
GalNAc-siRNA conjugates					
Givlaari, Givosiran (Alnylam Pharmaceuticals)	ALAS1	Acute hepatic porphyria	Subcutaneous	2019	575 k/year
Leqvio, inclisiran (Novartis/Alnylam Pharmaceuticals)	PCSK9	Hypercholesterolemia	Subcutaneous	2020	
Oxlumo, lumasiran (Alnylam Pharmaceuticals)	Glycolate oxidase	Primary hyperoxaluria type 1	Subcutaneous	2020	493 k/year
LNP-RNA					
Onpattro, patisiran (Alnylam Pharmaceuticals)	TTR siRNA	TTR-mediated amyloidosis	Intravenous	2018	450 k/year
Comirnaty, tozinameran (BioNTech/Pfizer)	SARS-CoV-2 spike protein mRNA	COVID-19 (FDA, emergency use; Switzerland, full approval)	Intramuscular	2020	30−40/dose
mRNA-1273 (Moderna/NIAID/BARDA)	SARS-CoV-2 spike protein mRNA	COVID-19 (FDA, emergency use)	Intramuscular	2020	30−36/dose
AAV vectors					
Glybera, alipogene tiparvovec (uniQure)	Lipoprotein lipase (LPL) (AAV1)	LPL deficiency	Intramuscular	2012 (withdrawn in 2017)	1 M
Luxturna, voretigene neparvovec-rzyl (Spark Therapeutics)	RPE65 (AAV2)	Leber congenital amaurosis	Subretinal	2017	850 k
Zolgensma, onasemnogene abeparvovec (AveXis/Novartis)	SMN1 (AAV9)	Spinal muscular atrophy	Intravenous	2019	2.1 M
Adenovirus (Ad) vectors					
Vaxzevria, AZD1222, ChAdOx1 nCoV-19 (AstraZeneca)	SARS-CoV-2 spike protein DNA(ChAdOx1)	COVID-19 (FDA, and EMA emergency use)	Intramuscular	2021	4−8/dose
Ad26.COV2.S (Johnson & Johnson)	SARS-CoV-2 spike protein DNA (Ad26)	COVID-19 (FDA, and EMA emergency use)	Intramuscular	2021	8.5−10/dose
Convidecia, Ad5-nCoV (CanSinoBIO)	SARS-CoV-2 spike protein DNA (Ad5)	COVID-19 (Approved in China)	Intramuscular	2021	30/dose

**Table 2 biomedicines-11-01275-t002:** A general plan for mRNA.

5′UTR cap
GAGAATAAACTAGTATTCTTCTGGTCCCCACAGACTCAGAGAGAACCCGCCACCATGT TCGTGTTCCTGGTGCTGCTGCCTCTGGTGTCCA
A start codon (Kozak)
GCAGCCAGTGCGTGAACCTGACCACCCGGACCCAGCTGCCACCAGCCTACACCAACAGCTTCA CCCGGGGCGTCTACTACCCCGACAAGGT
OPEN READING FRAME.
3′UTR
GCCCCTTTCCCGTCCTGGGTACCCCGAGTCTCCCCCGACCTCGGGTCCCAGGTATGCTCCC ACCTCCACCTGCCCCACTCACCACCTCTGCTAGTTCCAGACACCTCCCAAGCACGCAGCA ATGCAGCTCAAAACGCTTAGCCTAGCCACACCCCCACGGGAAACAGCAGTGATTAACCTT TAGCAATAAACGAAAGTTTAACTAAGCTATACTAACCCCAGGGTTGGTCAATTTCGTGCCAG CCACACCCTGGAGCTAGCA
poly(A) chain.

**Table 3 biomedicines-11-01275-t003:** Test methods used for ASOs, siRNAs, and mRNAs.

Category	Tests	ASOs	siRNAs	mRNAs
Active substance	Identification	Duplex melting temperature (UV absorbance)	Circular dichroism	Capillary gel electrophoresis
		FTIR	Duplex melting temperature (UV absorbance)	RT-Sanger sequencing
		IP-RPLC-UV-MS	FTIR	RT-PCR
		MS-MS	IP-RPLC UV-MS	
		NMR spectroscopy (^1^H, ^13^C and ^31^P)	MS-MS	
		X-ray diffraction	NMR spectroscopy (^1^H, ^13^C and ^31^P)	
			SEC-UV	
			UV spectroscopy	
	Assay	IP-RPLC-UV-MS	AEX-UV	UV spectroscopy
			UV spectroscopy	
	Impurities	IP-RPLC-UV-MS	AEX -UV	ddPCR
			IP-RPLC-UV-MS	Immunoblot
			SEC-UV	IP-RPLC
			2D-LC (AEX -UV (first dimension) and IP-RPLC-MS (second dimension))	qPCR
Finished medicinal product	Identification	Duplex melting temperature (UV spectroscopy)	AEX-UV	Capillary gel electrophoresis
		IP-RPLC-UV-MS	Duplex melting temperature (UV spectroscopy)	RT-Sanger sequencing
			FTIR	
			IP-RPLC-UV-MS	
	Assay	IP-RPLC-UV-MS	AEX-UV	AEX-UV
			UV spectroscopy	Fluorescence assay
	Impurities	IP-RPLC-UV-MS	AEX-UV	IP-RPLC-UV-MS
			IP-RPLC-UV-MS	

## Data Availability

Not applicable.

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
