# Peer review of "RNA Therapeutics: A Healthcare Paradigm Shift"

_biomedicines, 2023, doi:10.3390/biomedicines11051275_

Round 1
Reviewer 1 Report
The review provides a concise introduction on RNA therapeutics, focusing mostly on biotechnological aspects of the process and less on the rationale for target selection. The review is well-conceived and provides many useful information, facilitating access to original research. I have only some minor points of criticism, that I think will improve the manuscript.
- The "Testing" section (line 394) should not focus on immunization
- Table 3 could be omitted
- The "testing" section (line 687) should be reorganized, providing information on the different testing procedures depending on which property of the RNA is assayed. Maybe a table would be useful to summarize the methods used.
- In figure 2, S protein_mut could be replaced by CDS.
- Some references could be provided for: the comparison between dsRNA and ssRNA (line 111-113); Patisiran (line 113) and Givosiran (line 116); Cobomarsen, remlarsen, MRG-229, MRG-110 (lines 134-135) for these compounds please also check/rephrase the mode of action; the induction of apoptosis by doxorubicin induced by PANDA (line 146-147); tropism modification through engineered delivery systems (line 587-588).
English is of very high level, with only some editing required. Please see below:
- Please check the format in some citations, e.g. line 100, line 231, line 347, line 358, lines 453, 464, 465,, 471, 472, 473, 475, 477, 482
- Line 105: 3 should be switched to 3'
- Line 142: lnc RNA should be lncRNA
- Line 307-308: "third-party contract laboratories the desired DNA sequence is inserted by cloning" should be rephrased
- Line 333: MgCl2 should be MgCl2
- Line 360 and line 368: RNA purufucation and DNase I should be different sections/paragraph
- Line 383: Please rephrase "letting the mRNA gets destroyed by the abundance of RNA polymerases in the body"
- Line 396: "It involves inoculating e animals with the" should be rephrased to "It involves inoculating animals with the"
- Line 506: e "these lipids from the body..The development" should be "these lipids from the body. The development"
- Line 621: "kg1" should be "kg-1"
Author Response
I am thankful to the reviewer for taking the time and making suggestions that will significantly improve the quality of my submission. Please see below my response to all of your comments:
REVIEWER 1
The review provides a concise introduction on RNA therapeutics, focusing mostly on biotechnological aspects of the process and less on the rationale for target selection. The review is well-conceived and provides many useful information, facilitating access to original research. I have only some minor points of criticism, that I think will improve the manuscript.
- The "Testing" section (line 394) should not focus on immunization
FULLY REVISED AND ALL REFERENCE TO VACCINES REMOVED
- Table 3 could be omitted
TABLE 3 REMOVED
- The "testing" section (line 687) should be reorganized, providing information on the different testing procedures depending on which property of the RNA is assayed. Maybe a table would be useful to summarize the methods used.
REVISED AND A TABLE ADDED WITH SIMPLER EXPLANATION
- In figure 2, S protein_mut could be replaced by CDS.
NEW ARTWORK ADDED FOR MORE CLARIFICATION
- Some references could be provided for: the comparison between dsRNA and ssRNA (line 111-113); Patisiran (line 113) and Givosiran (line 116); Cobomarsen, remlarsen, MRG-229, MRG-110 (lines 134-135) for these compounds please also check/rephrase the mode of action; the induction of apoptosis by doxorubicin induced by PANDA (line 146-147); tropism modification through engineered delivery systems (line 587-588).
DONE
Comments on the Quality of English Language
English is of very high level, with only some editing required. Please see below:
- Please check the format in some citations, e.g. line 100, line 231, line 347, line 358, lines 453, 464, 465,, 471, 472, 473, 475, 477, 482
- Line 105: 3 should be switched to 3'
- Line 142: lnc RNA should be lncRNA
- Line 307-308: "third-party contract laboratories the desired DNA sequence is inserted by cloning" should be rephrased
- Line 333: MgCl2 should be MgCl2
- Line 360 and line 368: RNA purufucation and DNase I should be different sections/paragraph
done
- Line 383: Please rephrase "letting the mRNA gets destroyed by the abundance of RNA polymerases in the body"
DONE
- Line 396: "It involves inoculating e animals with the" should be rephrased to "It involves inoculating animals with the"
DONE
- Line 506: e "these lipids from the body..The development" should be "these lipids from the body. The development"
DONE
- Line 621: "kg1" should be "kg-1"
DONE
Reviewer 2 Report
In this paper, author provide a perspective on RNA technology and its applications.
Two should be addressed prior to publication.
(1) Add content about modified RNA on RNA Therapeutics.
It is very important, as Vaccines (e.g., for SARS-CoV2) used the modified RNA.
(2) Table 3 should be removed or shortened.
Author Response
I am thankful to the reviewer for making constructive suggestions.
In this paper, author provide a perspective on RNA technology and its applications.
Two should be addressed prior to publication.
(1) Add content about modified RNA on RNA Therapeutics.
It is very important, as Vaccines (e.g., for SARS-CoV2) used the modified RNA.
SECTION ADDED
(2) Table 3 should be removed or shortened.
REMOVED
Round 2
Reviewer 2 Report
This manuscript has been improved.